# Is mechanism of injury associated with outcome in spinal trauma? An observational cohort study from Tanzania

Chibuikem Anthony Ikwuegbuenyi[1,2], Julie Woodfield[1,2,3]*, François Waterkeyn[1,2,4], Scott L. Zuckerman[5], Beverly Cheserem[6], Andreas Leidinger[7], Albert Lazaro[2], Hamisi K. Shabani[2], Roger Härtl[1‡], Halinder S. Mangat[8‡]

1 Department of Neurological Surgery, New York Presbyterian Hospital/Och Spine, Weill Cornell Medicine, New York, New York, United States of America, 2 Division of Neurosurgery, Muhimbili Orthopedic and Neurosurgery Institute, Dar es Salaam, Tanzania, 3 Centre for Clinical Brain Sciences, University of Edinburgh, Edinburgh, United Kingdom, 4 Department of Neurosciences, Grand Hôpital de Charleroi, Charleroi, Belgium, 5 Department of Neurological Surgery, Vanderbilt University Medical Center, Nashville, Tennessee, United States of America, 6 Neurosurgery Unit, Aga Khan University Hospital, Nairobi, Kenya, 7 Department of Neurosurgery, Hospital de la Santa Creu i Sant Pau, Barcelona, Spain, 8 Department of Neurology, Kansas University Medical Center, Kansas City, Kansas, United States of America

‡ These authors are joint senior authors and contributed equally to this work
* julie.woodfield@ed.ac.uk

**Data Availability Statement:** The data are available within the manuscript. Any additional data

## Abstract

### Background

Traumatic spinal injury (TSI) is a disease of significant global health burden, particularly in low and middle-income countries where road traffic-related trauma is increasing. This study compared the demographics, injury patterns, and outcomes of TSI caused by road traffic accidents (RTAs) to non-traffic related TSI.

### Methods

A retrospective analysis was conducted using a neurotrauma registry from the Muhimbili Orthopaedic Institute (MOI) in Tanzania, a national referral center for spinal injuries. Patient sociodemographic characteristics, injury level, and severity were compared across mechanisms of injury. Neurological improvement, neurological deterioration, and mortality were compared between those sustaining TSI through an RTA versus non-RTA, using univariable and multivariable analyses.

### Results

A total of 626 patients were included, of which 302 (48%) were RTA-related. The median age was 34 years, and 532 (85%) were male. RTAs had a lower male preponderance compared to non-RTA causes (238/302, 79% vs. 294/324, 91%, p<0.001) and a higher proportion of cervical injuries (144/302, 48% vs. 122/324, 38%, p<0.001). No significant differences between RTA and non-RTA mechanisms were found in injury severity, time to admission, length of hospital stay, surgical intervention, neurological outcomes, or in-hospital mortality. Improved neurological outcomes were associated with incomplete injuries (AIS

supporting the research findings are available on request from the research co-ordinator at Muhimbili Orthopaedic Institute, Dar es Salaam, Tanzania for researchers who meet the criteria for access to confidential data. Please contact: moi. utafiti@gmail.com. Approval and consent for the database and its analysis does not allow transfer of the data outside of Tanzania or disclosure of any identifiable data. Due to the nature of the injuries described and the time period, releasing subject specific data risks identification of participants and would breach the agreements for analysis of the data.

**Funding:** The author(s) received no specific funding for this work.

**Competing interests:** The authors report no conflict of interest concerning the materials or methods used in this study or the findings specified in this paper. Scott L. Zuckerman is a consultant for the National Football League and Medtronic. Roger Hartl is a DePuy Synthes and Brainlab consultant and has royalties from Zimmer Biomet. He is also an advisor for 3D Bio and Real Spine. No other author declares any financial interests or personal relationships.

**Abbreviations:** AIS, American Spinal Injury Association Impairment Scale; CT, Computed Tomography; HDU, High dependency unit; IQR, interquartile range; LMICs, Low- and middle-income countries; MOI, Muhimbili Orthopedic Institute; MRI, Magnetic Resonance Imaging; NICU, Neuro-intensive care unit; RTAs, Road Traffic Accidents; STROBE, Strengthening the Reporting of Observational Studies in Epidemiology; TSI, Traumatic Spine Injuries; XR, X-ray.

B-D), while higher mortality rates were linked to cervical injuries and complete (AIS A) injuries.

## Conclusion

Our study in urban Tanzania finds no significant differences in outcomes between spinal injuries from road traffic accidents (RTAs) and non-RTA causes, suggesting the need for equitable resource allocation in spine trauma programs. Highlighting the critical link between cervical injuries and increased mortality, our findings call for targeted interventions across all causes of traumatic spinal injuries (TSI). We advocate for a comprehensive trauma care system that merges efficient pre-hospital care, specialized treatment, and prevention measures, aiming to enhance outcomes and ensure equity in trauma care in low- and middle-income countries.

## Introduction

Traumatic spinal injury (TSI) is a leading cause of death and disability globally, with an annual incidence of 45–80 per million population [1,2]. Survivors of spinal injuries may live with paralysis, pain and sensory changes, and bladder, bowel, and sexual dysfunction. Death and disability can lead to significant medical, social, psychological, and economic consequences for those affected and their families [3–5].

TSI occurs more frequently and has worse outcomes in low and middle-income countries than in high-income countries [1,2,6,7]. The higher prevalence of TSI may be due to the increased frequency of events causing injury, such as road traffic accidents (RTAs), and higher rates of occupational-related incidents, such as falls from trees [2,6,8]. Worse outcomes have been attributed to lower availability of first responders trained in resuscitation and spinal immobilization, delays and difficulties in accessing definitive care due to economic or infra-structure challenges, and lack of available spinal injuries acute care and rehabilitation services [1,6,7].

RTAs cause up to 50% of TSI worldwide [6], and the fatality rate overall from RTAs is as high as 27 per 100,000 people in the African continent [9–11]. Due to the significant impact of RTAs on global morbidity and mortality, the World Health Organization has advocated for a "Decade of Action for Road Safety" [12]. It is unknown whether spinal injuries resulting from RTAs have the same outcomes as those from other causes in low- and middle-income countries. This study aims to compare the injury characteristics and outcomes from TSI caused by RTA and TSI caused by non-RTA mechanisms of injury. This will identify the likely impact of targeted RTA prevention and management strategies.

## Methods

We report this observational cohort study in accordance with the Strengthening the Reporting of Observational Studies in Epidemiology (STROBE) guidelines [13].

### Study design and clinical setting

This is a retrospective analysis of a contemporaneous quality improvement neurotrauma regis-try. Between September 2016 and January 2022, consecutive patients with TSI were identified following hospital admission and included in the database [14]. For this study, the database

was assessed on the 30<sup>th</sup> of March 2023. The study was conducted at Muhimbili Orthopaedic Institute (MOI), the national neurosurgical and orthopedic referral center in Dar-es-Salaam, Tanzania, which the Tanzanian government runs. MOI has X-ray, Computed Tomography (CT), and Magnetic Resonance Imaging (MRI) facilities, high dependency and intensive care units (ICU), and orthopedic surgeons and neurosurgeons trained in the operative and peri-operative management of spinal injuries. There is limited provision of rehabilitation services. Patients are required to pay a fee for all services unless they have insurance. Spinal implants are available and are sometimes donated and free of charge to the patient [15]. At other times, the patient must purchase them, either out of pocket or through their insurance, before surgery.

The inclusion criteria for entry into the database were age over 14 years with an isolated TSI or TSI with concurrent mild traumatic brain injury (TBI), which is characterized by a Glasgow Coma Scale (GCS) score between 13 and 15, less than 24 hours of post-traumatic amnesia, and no significant intracranial injury on imaging. TSI included vertebral column fractures with or without neurological deficits and neurological deficits attributable to a spinal injury with or without bony injury. To establish a homogeneous study group conducive to our research objectives, we excluded patients with moderate to severe TBI (GCS $\leq$ 12, post-traumatic amnesia $\geq$ 24 hours, or requiring surgical intervention for intracranial injuries), polytrauma (multiple serious injuries in different body regions, excluding the spine), significant additional injuries (requiring surgical intervention, such as major long bone fractures, or significant thoracic, abdominal, or pelvic injuries), and those declared deceased upon arrival or before a comprehensive assessment could be completed. This exclusion strategy was designed to isolate the effects of spinal trauma, with or without mild TBI, on patient outcomes, thereby reducing the influence of confounding variables for a more precise analysis of recovery and outcomes. We acknowledge that trauma incidents often present complex injury patterns, yet our targeted approach allows for an in-depth examination within the specified patient population.

Participants were identified through a daily review of admitted patients. Routine clinical data were entered into the quality improvement registry by clinicians caring for the patients. The ethical review committees of the Muhimbili University of Health and Allied Sciences (Dar es Salaam, Tanzania) and Weill Cornell Medicine (New York, USA) determined the registry to be exempt from review as all data was directly anonymized in the Quality Improvement registry.

## Data collection and analysis

Database variables included age, sex, time to admission, level of injury, mechanism of injury, fracture type, imaging modality, American Spinal Injury Association Impairment Scale (AIS) [16] at admission and discharge, surgical intervention, time to surgery, length of hospital stay, and in-hospital mortality. The treating clinical team carried out the neurological assessments.

Patients in the database without a mechanism of injury recorded during the study period were excluded from this study. All other patients in the database were included in the analyses but excluded case-wise when missing data occurred. Mechanisms of injury were categorized into two groups: 1) RTAs, which included all injuries sustained by pedestrians, drivers, or passengers caused by motor vehicles, including motorcycles, and 2) non-RTAs, which included falls and other blunt or penetrating injuries. Change in AIS at discharge is any improvement or deterioration to a higher or lower grade. The level of injury was described as the highest spinal level where a fracture or the neurological deficit was present and were categorized as cervical or thoracolumbar. Frequency and percentage were reported for categorical variables. Age, time from injury to admission, and length of hospital stay were treated as non-parametric continuous variables and described using median and interquartile range (IQR).

To examine the relationship between the mechanism of injury and injury characteristics, sociodemographic data, improvement in AIS, deterioration in AIS, and in-hospital characteristics, the Mann-Whitney U test was used for continuous variables, and the Chi-square test for categorical variables. Logistic regression was performed to control for injury characteristics and sociodemographic data when assessing the relationship between the mechanism of injury and the outcome measures of improvement in AIS at discharge, deterioration in AIS at discharge, and in-hospital mortality. Those who were AIS E (neurologically intact) on admission were excluded from the outcome of neurological improvement as they could not improve. Those who were AIS A (complete injury) on admission were excluded from the outcome of deterioration as further neurological deterioration could not be analyzed. All analyses were done on STATA (StataCorp. 2013. Stata Statistical Software: Release 17. College Station, TX). Data visualization was done using R version 4.1.2 (Vienna, Austria).

## Results

### Mechanism of injury

Six hundred and twenty-nine patients with TSI were identified between September 2016 and January 2022. Three patients were excluded as the mechanism of injury was missing (Fig 1). Mechanisms of injury are shown in (Fig 2). RTAs (including all motorcycle, motor vehicle, and pedestrian injuries) accounted for 48.2% (302/626). Accidents involving motor vehicle occupants were the most common type of RTA leading to TSI (162/302, 53.6%), with motorcycle incidents accounting for 108/302 (35.7%) and accidents involving pedestrians occurring less frequently (32/302, 10.5%). The most common cause of TSI apart from RTAs was high-impact falls from a height greater than 3 meters (159/626, 25.2%).

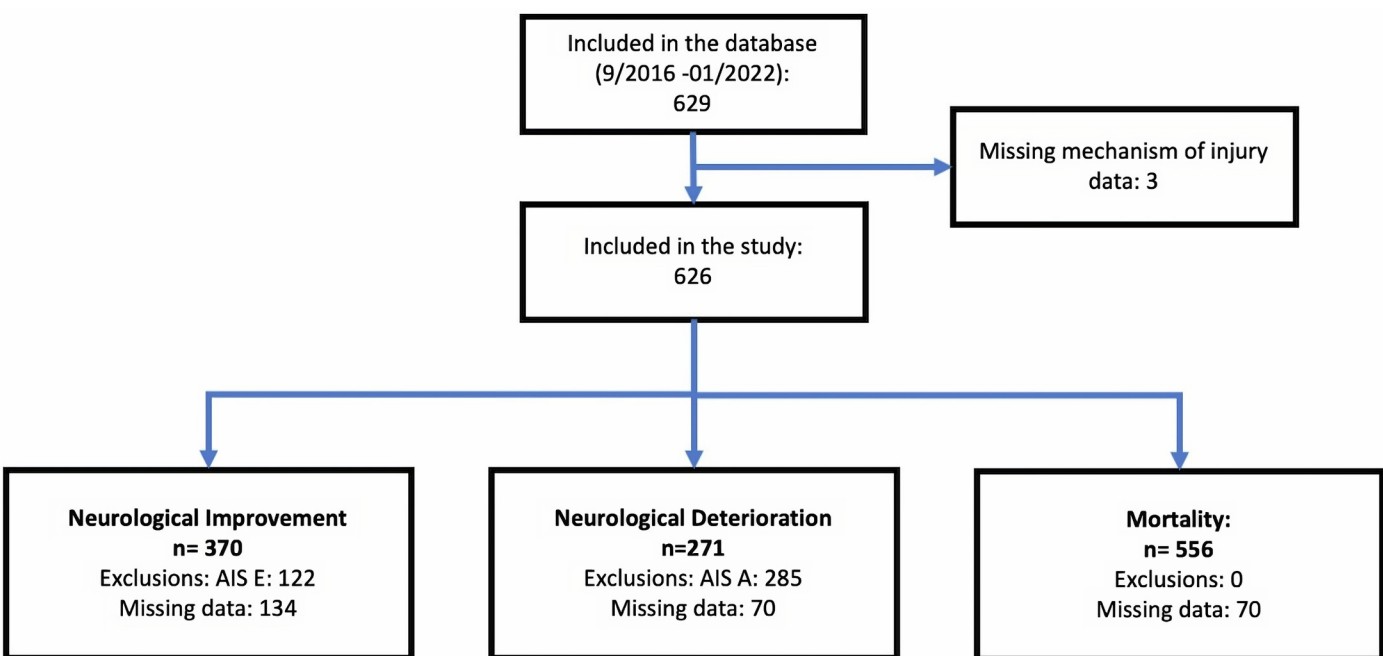

**Fig 1. Study flow diagram.** The number of patients included in the study and the outcome analyses. For outcome analyses, only complete cases were included. For neurological improvement, patients who were AIS E on admission were excluded. For neurological deterioration, patients who were AIS A on admission were excluded. (AIS: American Spinal Injury Association Impairment Scale).

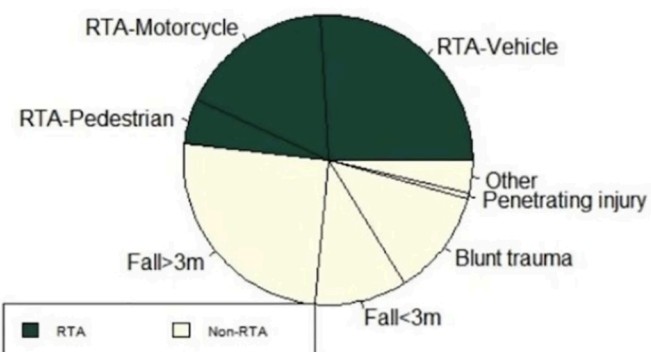

**Fig 2. Distribution of mechanism of injury.** Mechanisms of injury are grouped by RTA (shown in green) and non-RTA mechanisms (shown in yellow). A vehicle includes all driver or passenger accidents of motor vehicles; a motorcycle includes all driver or passenger accidents of motorcycles; a pedestrian includes all accidents where any motor vehicle/motorcycle injures pedestrians. High-impact falls (greater than 3m) are separated from low-impact falls (less than 3m). RTA: Road traffic accident. m: Meters.

## Demographics and management

The median age of those with TSI was 34 years (IQR 27–44), and the majority were male (n = 532/626, 85%) (Table 1). There was no difference in age between those who sustained a TSI in an RTA or non-RTA mechanism of injury (Table 1). However, spinal injuries caused by RTAs had a lower male preponderance than non-RTA spinal injuries (238/302, 79% vs. 294/324, 91%, p<0.001). Cervical injuries accounted for 266/626 (43%) overall, with a higher proportion sustained in RTAs compared to non-RTA mechanisms (144/302, 48%, vs. 122/324, 38%, p<0.001). AIS A accounted for nearly half of all injuries (285/626, 46%) and occurred with similar frequency due to RTA and non-RTA mechanisms of injury (130/302, 43% vs. 155/324, 48%, p = 0.42). Overall, 336/626 (54%) patients underwent surgery, and among the 336 who had surgery, 283 had information on the time to surgery available, with a median time of 13 days (IQR 5–27). There was no difference in surgical treatment, time to admission, or length of stay between RTA and non-RTA mechanisms of injury (Table 1).

Within this cohort of 626 patients, 24 had incomplete data regarding fracture type, and 5 showed no evidence of fractures on imaging. From the 597 patients with complete fracture data, 678 distinct fractures were identified. Spondylolisthesis Grades I and II were the most common, accounting for 37.46% of fractures (254/678), followed by burst/teardrop fractures at 33.33% (226/678). Other significant fracture types included compression fractures at 12.68% and cord contusions at 6.34%. Rarer findings included spondyloptosis and various C2 fractures, ranging from 0.15% to 1.33%.

Imaging was performed on all patients. X-rays were conducted on 83.5% (523 out of 626) of the patients. CT scans were performed on 31.9% (200 out of 626) of patients, and MRI scans were completed for 56.7% (355 out of 626) of the cohort.

Fig 3. shows the changes in AIS among the 508 patients who were alive at discharge and had AIS recorded at admission and discharge. 14/626 (2%) deteriorated, and 52/626 (8%) overall improved at discharge, with the majority, 442/626 (71%) remaining the same AIS at discharge (Table 1).

## Neurological improvement

Patients with AIS E at admission were excluded from this analysis (n = 122), and a further 134 had missing data (Fig 1), leaving 370 participants for analysis (Table 2). Of these, 47/370 (13%)

**Table 1. Demographics, injury patterns, and outcomes by the mechanism of injury.**

| Variables | Total n = 626 | Non-RTA n = 324 | RTA n = 302 | p-value |
|---|---|---|---|---|
| Age (years)* | 34 (27–44) | 35 (26–45) | 34 (27–43) | 0.658 |
| **Sex** | | | | <0.001 |
| Male | 532 (85.0) | 294 (90.7) | 238 (78.8) | |
| Female | 92 (14.7) | 28 (8.6) | 64 (21.2) | |
| Missing | 2 (0.3) | 2 (0.6) | 0 (0) | |
| Time to admission (days) † | 2 (1–6) | 2 (1–6) | 2 (1–6) | 0.849 |
| **Injury Level** | | | | 0.009 |
| Cervical | 266 (42.5) | 122 (37.7) | 144 (47.7) | |
| Thoracolumbar | 343 (54.8) | 196 (60.5) | 147 (48.7) | |
| Missing | 17 (2.7) | 6 (1.8) | 11 (3.6) | |
| **AIS on admission** | | | | 0.422 |
| A | 285 (45.5) | 155 (47.8) | 130 (43.1) | |
| B | 85 (13.6) | 48 (14.8) | 37 (12.3) | |
| C | 50 (8.0) | 24 (7.4) | 26 (8.6) | |
| D | 46 (7.4) | 20 (6.2) | 26 (8.6) | |
| E | 122 (19.5) | 56 (17.3) | 66 (21.9) | |
| Missing | 38 (6.1) | 21 (6.5) | 17 (5.6) | |
| **Surgical Treatment** | | | | 0.293 |
| Yes | 336 (53.7) | 168 (51.9) | 168 (55.6) | |
| No | 281 (44.9) | 153 (47.2) | 128 (42.4) | |
| Missing | 9 (1.4) | 3 (0.9) | 6 (2.0) | |
| Length of stay (days) ‡ | 22 (11–38) | 22 (11–41) | 21 (11–36) | 0.276 |
| **Change in AIS at discharge.** | | | | |
| Deterioration | 14 (2.2) | 12 (3.7) | 2 (0.7) | 0.059 |
| No Change | 442 (70.6) | 226 (69.8) | 216 (71.5) | |
| Improvement | 52 (8.3) | 29 (9.0) | 23 (7.6) | |
| Missing | 118 (18.8) | 57 (17.6) | 61 (20.2) | |
| **Mortality** | | | | 0.458 |
| Died | 94 (15.0) | 45 (13.9) | 49 (16.2) | |
| Alive | 515 (82.3) | 272 (84.0) | 243 (80.5) | |
| Missing | 17 (2.7) | 7 (2.2) | 10 (3.3) | |

RTA and non-RTA mechanisms of injury were compared using the Mann-Whitney U test for numeric variables and the Pearson chi-square test for categorical variables. Any improvement or deterioration in AIS to a different grade is shown. Patients of all AIS on admission are included. Data are median with IQR or count with percentage. Percentages include missing values. (RTA: Road traffic accident; IQR: Interquartile range; AIS: American Spinal Injury Association Impairment Scale;

*Two missing values;

† Three missing values;

‡Twenty missing values).

improved. Mechanism of injury was not associated with neurological improvement in either univariable or multivariable analyses (Table 2). However, those with incomplete injuries (AIS B–D) were more likely to improve compared to those with complete (AIS A) injuries in both univariable and multivariable analyses (Table 2).

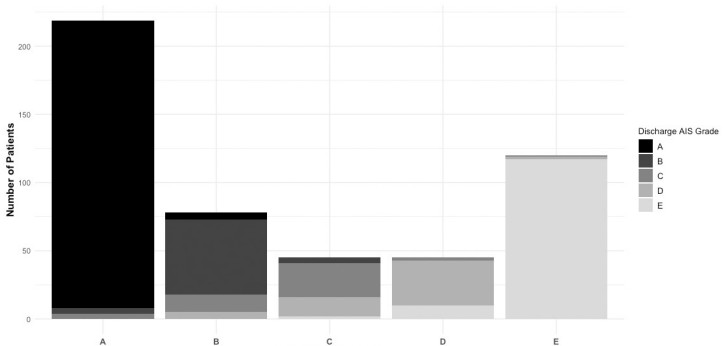

**Fig 3. Change in AIS from admission to discharge—stacked bar graph with AIS at admission on the x-axis.** We included the 508 patients who survived with AIS recorded at admission and discharge. The stacked bar graph is color-coded according to the AIS grade at discharge. The legend on the right shows the colors for AIS on discharge. AIS: American Spinal Injury Association Impairment Scale.

## Neurological deterioration

Patients with AIS A on admission (n = 285) and patients with missing data (n = 70) were excluded. Amongst those included, 13/271 (5%) deteriorated prior to discharge. Those sustaining TSI in an RTA were less likely to deteriorate than those who sustained a TSI through other

**Table 2. Mechanism of injury and neurological improvement.**

| Variables | No Improvement n = 323 | Improvement n = 47 | Unadjusted OR (95% CI) | p-value | Adjusted OR (95% CI) | p-value |
|---|---|---|---|---|---|---|
| **Mechanism of Injury** | | | | | | |
| RTA | 144 (44.6) | 20 (42.6) | 0.92 (0.50–1.71) | 0.794 | 0.78 (0.39–1.54) | 0.468 |
| Non-RTA | 179 (55.4) | 27 (57.5) | ref | | ref | |
| Age (years), median (IQR) | 32 (26–40) | 34 (27–46) | 1.02 (1.00–1.05) | 0.101 | 1.01 (0.99–1.04) | 0.323 |
| **Sex** | | | | | | |
| Male | 283 (87.6) | 38 (80.9) | 0.60 (0.27–1.33) | 0.202 | 0.71 (0.29–1.73) | 0.448 |
| Female | 40 (12.4) | 9 (19.2) | ref | | ref | |
| Time to admission (days), median (IQR) | 3 (1–7) | 2 (0–4) | 0.98 (0.95–1.02) | 0.356 | 0.98 (0.95–1.03) | 0.462 |
| **Injury Level** | | | | | | |
| Cervical | 114 (35.3) | 20 (42.6) | 1.36 (0.73–2.53) | 0.335 | 1.09 (0.53–2.21) | 0.815 |
| Thoracolumbar | 209 (64.7) | 27 (57.5) | ref | | ref | |
| **AIS Grade on admission** | | | | | | |
| A | 204 (63.2) | 8 (17.0) | ref | | ref | |
| B | 59 (18.3) | 17 (36.2) | 7.35 (3.02–17.87) | <0.001 | 6.83 (2.78–16.79) | <0.001 |
| C | 26 (8.1) | 14 (29.8) | 13.73 (5.26–35.85) | <0.001 | 14.28 (5.26–38.73) | <0.001 |
| D | 34 (10.5) | 8 (17.0) | 6.00 (2.11–17.06) | 0.001 | 6.27 (2.14–18.33) | 0.001 |
| **Surgical Treatment** | | | | | | |
| Yes | 198 (61.3) | 34 (72.3) | 1.65 (0.84–3.26) | 0.144 | 1.94 (0.90–4.17) | 0.091 |
| No | 125 (38.7) | 13 (27.7) | ref | | ref | |

Association between the mechanism of injury and neurological improvement. A total of 370 participants were included. Patients who were AIS E on admission were excluded (n = 122). Data are counted (%) or median with IQR. ORs are shown with 95% CI. (RTA; road traffic accident; OR: Odds ratio; AIS: American Spinal Injury Association Impairment Scale; IQR: Interquartile range; CI: Confidence interval).

**Table 3. Mechanism of injury and neurological deterioration.**

| Variables | No deterioration = 258 | Deterioration n = 13 | Unadjusted OR (95% CI) | p-value | Adjusted OR (95% CI) | p-value |
|---|---|---|---|---|---|---|
| **Mechanism of Injury** | | | | | | |
| RTA | 134 (51.9) | 2 (15.4) | 0.17 (0.04–0.79) | 0.010 | 0.13 (0.03–0.66) | 0.014 |
| Non-RTA | 124 (48.1) | 11 (84.6) | Ref | | ref | |
| Age (years), median (IQR) | 35 (28–45) | 30 (23–47) | 0.99 (0.94–1.04) | 0.636 | 0.98 (0.94–1.03) | 0.499 |
| **Sex** | | | | | | |
| Male | 213 (82.6) | 10 (76.9) | 0.70 (0.19–2.67) | 0.604 | 0.40 (0.09–1.76) | 0.226 |
| Female | 45 (17.4) | 3 (23.1) | ref | | Ref | |
| Time to admission (days), median (IQR) | 2 (1–6) | 3 (1–6) | 1.00 (0.96–1.04) | 0.886 | 1.01 (0.96–1.05) | 0.786 |
| **Injury Level** | | | | | | |
| Cervical | 90 (34.9) | 7 (53.9) | 2.18 (0.71–6.68) | 0.173 | 2.62 (0.76–9.09) | 0.129 |
| Thoracolumbar | 168 (65.1) | 6 (46.2) | ref | | Ref | |
| **AIS Grade on admission** | | | | | | |
| B | 71 (27.5) | 5 (38.5) | 2.58 (0.60–11.1) | 0.204 | 2.04 (0.44–9.59) | 0.365 |
| C | 37 (14.3) | 3 (23.1) | 2.97 (0.57–15.4) | 0.194 | 1.60 (0.26–9.94) | 0.617 |
| D | 40 (15.5) | 2 (15.4) | 1.83 (0.30–11.4) | 0.515 | 1.37 (0.20–9.54) | 0.750 |
| E | 110 (42.6) | 3 (23.1) | ref | | ref | |
| **Surgical Treatment** | | | | | | |
| Yes | 146 (56.6) | 7 (53.9) | 0.89 (0.29–2.74) | 0.846 | 0.85 (0.25–2.87) | 0.793 |
| No | 112 (43.4) | 6 (46.2) | ref | | ref | |

The association between the mechanism of injury and neurological deterioration at discharge in AIS among those who survived. 271 patients with complete data were included in the analysis. Patients who were AIS A on admission were excluded. Data are count (percentage) or median (IQR). ORs are shown with 95% CIs. (RTA: Road traffic accident; OR: Odds ratio; AIS: American Spinal Injury Association Impairment Scale; 95%CI: Confidence interval).

mechanisms of injury in both the univariable (2/13, 15% vs. 11/13, 85%, p = 0.010) and multivariable analyses (p = 0.014) (Table 3). Neurological deterioration was not associated with age, sex, time to admission, injury level, AIS on admission, or surgical treatment (Table 3).

## Mortality

94/626 (15%) of all patients with TSI died during admission (Table 1), and 556 patients had complete data for analysis (Fig 1). Table 4 compares those known to have died as inpatients with those who were alive at discharge. There was no difference in mortality between RTA and non-RTA mechanisms of injury in either univariable or multivariable analyses (Table 4). Mortality was associated with cervical injury, with 62/66 (94%) of those dying sustaining a cervical injury, a slightly older age (median: 38 years vs median 34 years), and not undergoing surgery (Table 4). Of those who died, 24/66 (36%) underwent surgery, whereas 290/490 (60%) of survivors underwent surgery. None of those who were AIS E or D died, and those who were AIS B or C on admission were less likely to die than those who were AIS A (Table 4).

## Discussion

TSI caused by RTAs in Tanzania is more likely to result in cervical injury but less likely to be associated with neurological deterioration during admission. Although few women sustain TSI overall in Tanzania, the RTA mechanism of injury is more common for women than a non-RTA mechanism of injury. RTA is more likely to result in cervical injury, and cervical injury is more likely to result in death. Still, there was no independent association in

**Table 4. Mechanism of injury and mortality.**

| Variables | Alive n = 490 | Died N = 66 | unadjusted OR (95% CI) | p-value | adjusted OR (95% CI) | p-value |
|---|---|---|---|---|---|---|
| **Mechanism of Injury** | | | | | | |
| RTA | 229 (46.7) | 37 (56.1) | 1.45 (0.87–2.44) | 0.155 | 1.56 (0.81–3.01) | 0.182 |
| Non-RTA | 261 (53.3) | 29 (43.9) | Ref | | ref | |
| **Age** (years) | 34 (26–42) | 38 (29–47) | 1.02 (1.00–1.04) | 0.048 | 1.03 (1.00–1.06) | 0.023 |
| **Sex** | | | | | | |
| Male | 415 (84.7) | 55 (83.3) | 0.90 (0.45–1.81) | 0.774 | 0.69 (0.27–1.71) | 0.417 |
| Female | 75 (15.3) | 11 (16.7) | ref | | ref | |
| **Time to admission** | 3 (1–6) | 2 (1–4) | 0.91 (0.85–0.98) | 0.011 | 0.92 (0.85–1.00) | 0.065 |
| **Injury Level** | | | | | | |
| Cervical | 165 (33.7) | 62 (93.9) | 30.5 (10.9–85.4) | <0.001 | 24.9 (8.58–72.1) | <0.001 |
| Thoracolumbar | 325 (66.3) | 4 (6.1) | ref | | ref | |
| **AIS Grade on admission** | | | | | | |
| A | 215 (43.8) | 58 (87.9) | ref | | ref | |
| B | 78 (15.9) | 4 (6.1) | 0.19 (0.07–0.54) | 0.002 | 0.13 (0.04–0.41) | 0.001 |
| C | 41 (8.4) | 4 (6.1) | 0.36 (0.12–1.05) | 0.062 | 0.16 (0.05–0.52) | 0.002 |
| D | 42 (8.6) | 0* | | | | |
| E | 114 (23.3) | 0* | | | | |
| **Surgery** | | | | | | |
| Yes | 290 (59.2) | 24 (36.4) | 0.39 (0.23–0.68) | 0.001 | 0.46 (0.23–0.91) | 0.025 |
| No | 200 (40.8) | 42 (63.6) | ref | | ref | |

Association between mechanism of injury and mortality. There were 556 patients without missing data included. In the multivariable analysis and the comparison of AIS grade only 400 patients were included as AIS grades D and E were excluded because there were no patients of AIS grade D or E on admission who died. Data are count (percentage) or median (IQR). ORs are shown with 95% CIs. (RTA: Road traffic accident; OR: Odds ratio; AIS: American Spinal Injury Association Impairment Scale; CI: Confidence interval; IQR: Interquartile range).

multivariable analysis between the mechanism of injury and death, suggesting cervical injury is related to mortality regardless of mechanism. Patients sustaining complete (AIS A) injuries are less likely to show neurological improvement and more likely to die in hospital, but in this cohort, sustaining an AIS A injury was just as likely in non-RTA mechanisms as in an RTA.

These data highlight the need for public health prevention and first responder management strategies to equally address non-RTA related TSI due to falls from height and other mechanisms of injury along with RTAs to reduce incidence and improve outcomes for TSI in low and middle-income countries.

The study's strengths include the large sample size (n = 626) from a database collected over five years, with the same data points consistently recorded. However, not all patients in the database could be included due to missing data (see Table 1 and Fig 1). As only a few patients deteriorated or improved in AIS at discharge, the power of the statistical analysis is limited. This is in keeping with other reports of only a few patients improving or deteriorating following spinal injuries [17]. We only included data from a single institution, but MOI is the referral center for spine trauma, attracting patients from an average distance of approximately 280 kilometers [14], and there are only a few other centers in Tanzania offering specialist care for TSI. Therefore, the data represents those making it to the referral center alive following TSI. Because we combined non-RTA mechanisms of injury, we cannot comment on any differences between these, for example, between falls greater or less than 3m in height.

There is a significant gender gap in TSI, with 85% of those included being male. This is comparable to a systematic review of spinal cord injuries in developing countries, where 82.8% of TSI occurred in males [18]. We found a higher proportion of females involved in RTAs compared to non-RTAs. This could be due to lower female participation in activities such as tree climbing. A recent study investigating spine trauma incidents resulting from coconut tree falls reported that all patients during the study period were males [8]. The male preponderance for TSI overall highlights the target group for preventative strategies for both RTA and non-RTA mechanisms of injury. The primary causes of TSI remain consistent with previous studies with RTAs and high-impact falls as the main causes [2,18].

Cervical and complete (AIS A) injuries were associated with very high mortality, and those with AIS A were less likely to show neurological improvement. Although cervical injuries were more common in RTAs than other mechanisms of injury, there was no independent association between mechanism and inpatient death or neurological improvement, suggesting that once the injury has occurred, the level and severity of the injury rather than the mechanism determine outcome [19]. There was an association between the RTA mechanism of injury and less likelihood of neurological deterioration as an inpatient. However, only a few patients showed deterioration as an inpatient, so this must be interpreted with caution. As the deterioration was measured from admission to discharge, this cannot be due to different pre-hospital care for different mechanisms of injury. Surgical treatment was associated with lower mortality, but it is unclear if this is due to surgical treatment only being carried out in those likely to survive or who did survive the first few days to weeks in hospital.

As outcomes were similar between RTA and non-RTA mechanisms of injury, all mechanisms of injury should be targeted for preventative public health measures and improving trauma care systems in relation to spinal trauma [20]. Strengthening trauma systems could include measures such as providing prehospital care, implementing triage criteria for transportation to ensure timely and appropriate care, particularly for cervical spine injuries, and establishing designated trauma centers. A recent study highlights the potential impact of implementing a comprehensive trauma system with 100% coverage in LMICs, estimating that it could save over 200,000 lives annually on a global scale [7], even with partial improvements in the system, such as the establishment of trauma centers, approximately 145,000 lives could be saved each year [7].

Strengthening trauma systems in the region will be challenging, requiring a comprehensive approach that involves research, policy, infrastructure, workforce, collaboration, and achieving universal health coverage [20–22]. African countries can significantly enhance their health systems and improve population health outcomes by investing in health policy research, developing robust national health programs, addressing infrastructure issues, fostering collaboration, and striving for universal health coverage. This is particularly important for conditions such as TSI, which have a significant social, psychological, medical, and economic impact on young adults and their families.

Preventive strategies could potentially reduce spine trauma in the region by half [23]. Recent research suggests the utilization of strong enforcement initiatives and legislative reforms has yielded the best results in reducing incidents of RTAs [24]. Such interventions include enforcing seatbelt usage [25,26], mandating helmets for motorcycle and bicycle riders [27–29], traffic enforcement, and implementing traffic-calming measures, including speed bumps [30–32]. These interventions have been shown to be effective in decreasing road traffic accidents in high-income countries [33–35]. Furthermore, the adoption and implementation of global initiatives such as the Decade of Action for Road Safety 2011–2020 [36] and the UN Sustainable Development Goals (SDGs) targeting a 50% reduction in road traffic deaths by the year 2020 [37] has led to the reduction of road traffic crashes and associated injuries.

Similar initiatives on a global scale are required to prevent spine trauma from other mechanisms like falls from heights. On an individual level, employing safety measures such as using appropriate personal protective equipment, like helmets and harnesses, when working at heights is crucial [8]. Proper training and adherence to safety protocols can significantly reduce the risk of falls. Regular equipment inspections and maintenance ensure that safety devices are in optimal condition. Employing guardrails, safety nets, and toe boards on construction sites or elevated work areas can provide an added layer of protection. When engaging in recreational activities like rock climbing, tree trimming, or harvesting, following safety guidelines and receiving adequate training are imperative to mitigate the risk of falls and subsequent spine injuries. Overall, a combination of preparedness, proper equipment, and adherence to safety standards is essential in preventing spine injuries resulting from falls from heights.

Public health messaging and education of dangers posed by typically 'safe' activities and preventative strategies is important. Education and awareness campaigns may be pivotal in addressing the complex issue of spinal injuries [38]. First, these campaigns are powerful tools to enlighten the public about such injuries' profound and often life-altering consequences. Spinal trauma can devastate individuals, affecting their physical and emotional well-being, often leading to long-term disabilities and significantly impeding their quality of life. These repercussions extend beyond the individual, as the social and economic burden on families and communities can be substantial. Understanding the full scope of these consequences is fundamental to generating empathy and a sense of urgency in tackling this critical public health concern.

In addition to conveying the gravity of spinal injuries, education and awareness campaigns also focus on the importance of prevention. They shed light on the multifaceted causative mechanisms that give rise to these injuries. Whether resulting from road traffic accidents, falls, sports-related incidents, workplace hazards, or various recreational activities, the awareness initiatives aim to impart knowledge about safety measures and best practices. This can encompass advocating for using seat belts and helmets, responsible and defensive driving behaviors, safe working conditions, and implementing effective fall prevention strategies. By disseminating information on these preventative measures, these campaigns empower individuals, communities, and organizations with the tools and understanding necessary to minimize the risks of spinal injuries.

Furthermore, the reach of these educational and awareness campaigns is not confined to a singular level of implementation. Instead, they are adaptable and versatile, capable of being launched at various strata of society. Initiatives can take root at the grassroots community level, where individuals can address local issues and concerns with cultural sensitivity and localized strategies. Moving up, campaigns at the local and national levels enable governments, healthcare organizations, and community leaders to collaborate in implementing policies, regulations, and interventions to reduce spinal injuries. On a broader scale, continental and global campaigns foster international cooperation, facilitating the sharing of knowledge, best practices, and the allocation of resources to combat spinal injury prevention on a larger stage.

## Conclusion

Our thorough analysis of the urban landscape in Tanzania has revealed that there are no significant differences in outcomes between spinal injuries caused by road traffic accidents (RTAs) and non-RTA sources. This finding shows that it is important to allocate resources fairly and equally within spine trauma programs. Prevention and treatment strategies should receive the same amount of attention, regardless of the cause of the spinal injury. Our study challenges the

traditional allocation of resources, which typically prioritizes RTAs. Instead, we recommend that efforts should be extended to encompass non-RTA-related spinal injuries equally.

The study emphasizes the crucial connection between cervical injuries and an increased risk of mortality. This highlights the necessity for targeted interventions, regardless of the injury mechanism. Additionally, the study revealed gender disparities in TSI (traumatic spinal injuries) incidence, indicating the need for tailored public health strategies that consider societal roles and activities. Ultimately, the research supports the establishment of a holistic trauma care system that integrates efficient pre-hospital care, specialized treatment, and robust public health prevention measures. This system can significantly reduce the burden of TSI in low- and middle-income countries, leading to improved outcomes and equity in trauma care.

## Acknowledgments

We thank the interdisciplinary team at the Muhimbili Orthopedic Institute, including the nurses, residents, fellows, and specialists involved in these patients' care.

## Author Contributions

**Conceptualization:** Chibuikem Anthony Ikwuegbuenyi, Julie Woodfield, Scott L. Zuckerman, Roger Härtl, Halinder S. Mangat.

**Data curation:** Chibuikem Anthony Ikwuegbuenyi, François Waterkeyn.

**Formal analysis:** Chibuikem Anthony Ikwuegbuenyi, Julie Woodfield, Halinder S. Mangat.

**Funding acquisition:** Roger Härtl.

**Investigation:** Chibuikem Anthony Ikwuegbuenyi.

**Methodology:** Chibuikem Anthony Ikwuegbuenyi, Julie Woodfield, François Waterkeyn, Beverly Cheserem, Andreas Leidinger, Albert Lazaro, Hamisi K. Shabani, Roger Härtl, Halinder S. Mangat.

**Resources:** Chibuikem Anthony Ikwuegbuenyi.

**Software:** Chibuikem Anthony Ikwuegbuenyi.

**Supervision:** Julie Woodfield, Roger Härtl.

**Validation:** Roger Härtl, Halinder S. Mangat.

**Visualization:** Chibuikem Anthony Ikwuegbuenyi, Julie Woodfield.

**Writing – original draft:** Chibuikem Anthony Ikwuegbuenyi, Julie Woodfield, François Waterkeyn, Scott L. Zuckerman, Beverly Cheserem, Andreas Leidinger, Albert Lazaro, Hamisi K. Shabani, Roger Härtl, Halinder S. Mangat.

**Writing – review & editing:** Chibuikem Anthony Ikwuegbuenyi, Julie Woodfield, François Waterkeyn, Scott L. Zuckerman, Beverly Cheserem, Andreas Leidinger, Albert Lazaro, Hamisi K. Shabani, Roger Härtl, Halinder S. Mangat.

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
