## [Decision Letter · Decision Letter 0]

1 Feb 2024

PONE-D-23-38938Is Mechanism of Injury Associated with Outcome in Spinal Trauma? An Observational Cohort Study from TanzaniaPLOS ONE

Dear Dr. Woodfield,

Thank you for submitting your manuscript to PLOS ONE. After careful consideration, we feel that it has merit but does not fully meet PLOS ONE’s publication criteria as it currently stands. Therefore, we invite you to submit a revised version of the manuscript that addresses the points raised during the review process.

We look forward to receiving your revised manuscript.

Kind regards,

Alvan Ukachukwu, MD, MSc.GH

Academic Editor

PLOS ONE

Journal Requirements:

The authors report no conflict of interest concerning the materials or methods used in this study or the findings specified in this paper. Roger Hartl declares consulting work for DePuy Synthes, Brainlab, and Ulrich. Roger Hartl reports a financial relationship with Zimmer Biomet and Real Spine. No other author declares any financial interests or personal relationships.

We note that one or more of the authors are employed by a commercial company.

“The funder provided support in the form of salaries for authors, but did not have any additional role in the study design, data collection and analysis, decision to publish, or preparation of the manuscript. The specific roles of these authors are articulated in the ‘author contributions’ section.”

3. In the online submission form you indicate that your data is not available for proprietary reasons and have provided a contact point for accessing this data. Please note that your current contact point is a co-author on this manuscript. According to our Data Policy, the contact point must not be an author on the manuscript and must be an institutional contact, ideally not an individual. Please revise your data statement to a non-author institutional point of contact, such as a data access or ethics committee, and send this to us via return email. Please also include contact information for the third party organization, and please include the full citation of where the data can be found.

Additional Editor Comments:

The reviewers appreciate the authors' efforts to characterize outcomes of traumatic spina injury in Tanzania, a low income country. The discussion on prevention was particularly interesting. However, the reviewers raise salient points which, if addressed, will strengthen the work done by the authors. The authors are requested to revise their manuscript accordingly and address all reviewers' comments.

Reviewers' comments:

Reviewer's Responses to Questions

**Comments to the Author**

1. Is the manuscript technically sound, and do the data support the conclusions?

Reviewer #1: Yes

Reviewer #2: Partly

2. Has the statistical analysis been performed appropriately and rigorously? 

Reviewer #1: Yes

Reviewer #2: Yes

3. Have the authors made all data underlying the findings in their manuscript fully available?

Reviewer #1: Yes

Reviewer #2: Yes

4. Is the manuscript presented in an intelligible fashion and written in standard English?

Reviewer #1: Yes

Reviewer #2: Yes

5. Review Comments to the Author

Reviewer #1: Dear Author,

I recently read your research paper and I wanted to express my appreciation for your work. The clarity of your writing, coupled with the thorough literature review impressed me. Your research adds valuable insights to the the mechanism of spinal injury and outcome in low income countries, and I look forward to seeing its impact.

Thank you for your contribution to the academic community.

Reviewer #2: The authors, in the "Study design and clinical setting" section, define the inclusion/exclusion criteria. Patients with "concomitant moderate or severe traumatic brain injury, polytrauma, additional injuries such as long bone fractures, chest, abdominal, or pelvic injury, and mortality" are excluded. This definition should be better defined. It seems difficult to me to believe that patients involved in road accidents or falls from considerable heights (3 metres) only report spinal trauma.

The authors do not specify either the type of traumatic injury or the type of imaging performed on the patients (vertebral dislocation, burst fracture, hematoma).

The authors do not report the timing of surgical treatment.

Both points highlighted above affect the outcome of such patients.

The conclusions are not incisive. The greater severity and higher incidence of mortality are known to be more frequent in patients suffering from trauma in the cervical region.

The final part of the discussion in which the authors discuss prevention appears very interesting.

6. PLOS authors have the option to publish the peer review history of their article (what does this mean?). If published, this will include your full peer review and any attached files.

Reviewer #1: **Yes: **Yared Nigusie Abebe

Reviewer #2: **Yes: **Gerardo Caruso

---

## [Author Response · Author response to Decision Letter 0]

6 Jun 2024

Ref.: Ms. No. PONE-D-23-38938

Is Mechanism of Injury Associated with Outcome in Spinal Trauma? An Observational Cohort Study from Tanzania

Dear Dr. Emily Chenette, Editor & Reviewers,

Thank you for your time and efforts. Please find our revised manuscript attached, in which we implemented and addressed the reviewers' requests and concerns. We have made the utmost effort to address all reviewers’ comments thoroughly, and the following are point-by-point responses to each reviewer. 

Editor: 

Response: 

Thank you for your comment. We have adjusted sections to meet these formatting guidelines. 

Editor: 

The authors report no conflict of interest concerning the materials or methods used in this study or the findings specified in this paper. Roger Hartl declares consulting work for DePuy Synthes, Brainlab, and Ulrich. Roger Hartl reports a financial relationship with Zimmer Biomet and Real Spine. No other author declares any financial interests or personal relationships.

We note that one or more of the authors are employed by a commercial company.

“The funder provided support in the form of salaries for authors, but did not have any additional role in the study design, data collection and analysis, decision to publish, or preparation of the manuscript. The specific roles of these authors are articulated in the ‘author contributions’ section.”

Response:

Thank you for your comment. We wish to clarify two important points:

1. No Commercial Employment: None of the authors are employed by a commercial company. Our manuscript has transparently disclosed all affiliations and potential conflicts of interest. However, we would like to clarify that the implants used in managing these patients were, at some point, kindly donated by DePuy. This donation did not influence the study design, data collection and analysis, publication decision, or manuscript preparation.

2. No Funding: This study received no external funding, including no financial support for salaries or research materials from any commercial or non-commercial entities.

Given these clarifications, we have written the following amendments:

Competing Interests: “The authors report no conflict of interest concerning the materials or methods used in this study or the findings specified in this paper. Scott L. Zuckerman is a consultant for the National Football League and Medtronic. Roger Hartl is a DePuy Synthes and Brainlab consultant and has royalties from Zimmer Biomet. He is also an advisor for 3D Bio and Real Spine. No other author declares any financial interests or personal relationships.”

Funding Statement: "This research received no specific grant from funding agencies in the public, commercial, or not-for-profit sectors. While this research did not receive specific funding, DePuy Synthes donated some implants used in the management of patients in this study. This support did not influence any aspect of the study.”

We have revised our manuscript to reflect these points and ensure accurate transparency.

Editor: 

3. In the online submission form you indicate that your data is not available for proprietary reasons and have provided a contact point for accessing this data. Please note that your current contact point is a co-author on this manuscript. According to our Data Policy, the contact point must not be an author on the manuscript and must be an institutional contact, ideally not an individual. Please revise your data statement to a non-author institutional point of contact, such as a data access or ethics committee, and send this to us via return email. Please also include contact information for the third party organization, and please include the full citation of where the data can be found.

Response:

Thank you for your comment. We have adjusted this accordingly. 

Editor: 

Response:

Thank you for your comment. Our reference list is complete and correct and includes no retracted papers. 

Reviewer #1: 

Dear Author,

I recently read your research paper and I wanted to express my appreciation for your work. The clarity of your writing, coupled with the thorough literature review impressed me. Your research adds valuable insights to the the mechanism of spinal injury and outcome in low income countries, and I look forward to seeing its impact.

Thank you for your contribution to the academic community.

Response: 

Thank you for your kind words and encouragement regarding our manuscript on spinal injury mechanisms and outcomes in low-income countries. We are pleased to hear that our work resonated with you and appreciate your recognition of its potential impact. We look forward to contributing further to this field.

Reviewer #2: The authors, in the "Study design and clinical setting" section, define the inclusion/exclusion criteria. Patients with "concomitant moderate or severe traumatic brain injury, polytrauma, additional injuries such as long bone fractures, chest, abdominal, or pelvic injury, and mortality" are excluded. This definition should be better defined. It seems difficult to me to believe that patients involved in road accidents or falls from considerable heights (3 metres) only report spinal trauma.

Response: 

Thank you for your insightful comments and for raising an important point regarding the definition and plausibility of our inclusion/exclusion criteria, particularly concerning patients with only spinal trauma versus those with multiple injuries in scenarios such as road accidents or falls from considerable heights.

To address your concern, we have revised our manuscript's "Study design and clinical setting" section to provide a more detailed explanation of our inclusion and exclusion criteria. Specifically, we have clarified the rationale behind focusing on patients with isolated spinal trauma or those with spinal trauma and concurrent mild TBI, and we have precisely defined these conditions. We have also refined our exclusion criteria to explicitly detail the types of injuries and conditions that led to exclusion from the study. Here is a brief overview of the modifications made:

1. Inclusion Criteria Clarification: We have specified that our study focuses on patients presenting with isolated spinal trauma or spinal trauma with concurrent mild TBI, characterized by specific Glasgow Coma Scale scores and other parameters.

2. Exclusion Criteria Refinement: We have clearly defined moderate to severe TBI, polytrauma, additional significant injuries requiring surgical intervention, and mortality at presentation. This refinement aims to ensure the study population is realistically representative of spinal trauma patients, addressing the feasibility of encountering patients with only spinal trauma.

3. Rationale for Exclusion Criteria: We have added a statement acknowledging the complexity of injury patterns in trauma incidents but justified our focused approach to allow for an in-depth examination of spinal trauma outcomes within our defined patient population.

These revisions directly address your concerns by enhancing the clarity and plausibility of our study design and patient selection criteria. We hope these changes adequately respond to your feedback and strengthen the manuscript.

Reviewer #2: 

The authors do not specify either the type of traumatic injury or the type of imaging performed on the patients (vertebral dislocation, burst fracture, hematoma).

The authors do not report the timing of surgical treatment.

Both points highlighted above affect the outcome of such patients.

Response: 

Thank you for your valuable feedback. We have now included the types of fractures observed and the specific imaging modalities used. We acknowledge that while the primary aim was to investigate the mechanism of injury, detailing the fracture types and imaging adds important context and has been incorporated accordingly.

Additionally, we have provided data on the timing of surgical interventions for the 336 patients who underwent surgery. We found that the median time to surgery for the 283 patients with available data was 13 days (IQR 5-27). This information is now clearly reported in the revised manuscript, enhancing the understanding of treatment timelines and their potential impact on outcomes.

These amendments address your concerns and enrich the manuscript. Thank you for guiding the improvement of our study.

Reviewer #2: 

The conclusions are not incisive. The greater severity and higher incidence of mortality are known to be more frequent in patients suffering from trauma in the cervical region.

Response: 

Thank you for your constructive feedback on our conclusions. We have taken your suggestions into account and revised our conclusions. We have sharpened our focus to highlight the key findings, especially regarding the similar outcomes of spinal injuries from RTA and non-RTA sources and the critical importance of addressing cervical injuries due to their higher mortality risk. We have also emphasized the necessity of equitable resource allocation in spine trauma programs and the implementation of targeted interventions. These revisions aim to provide a clearer and more incisive synthesis of our research implications, aligning with your valuable suggestions.

Reviewer #2:

The final part of the discussion in which the authors discuss prevention appears very interesting.

Response: 

Thank you for your positive feedback regarding our discussion on prevention. We are glad you found it interesting and value your encouragement.

---

## [Decision Letter · Decision Letter 1]

20 Jun 2024

Is Mechanism of Injury Associated with Outcome in Spinal Trauma? An Observational Cohort Study from Tanzania

PONE-D-23-38938R1

Dear Dr. Julie Woodfield,

We’re pleased to inform you that your manuscript has been judged scientifically suitable for publication and will be formally accepted for publication once it meets all outstanding technical requirements.

Kind regards,

Alvan-Emeka K. Ukachukwu, MD, MSc.GH

Academic Editor

PLOS ONE

Additional Editor Comments (optional):

Reviewers' comments:

Reviewer's Responses to Questions

**Comments to the Author**

1. If the authors have adequately addressed your comments raised in a previous round of review and you feel that this manuscript is now acceptable for publication, you may indicate that here to bypass the “Comments to the Author” section, enter your conflict of interest statement in the “Confidential to Editor” section, and submit your "Accept" recommendation.

Reviewer #2: All comments have been addressed

2. Is the manuscript technically sound, and do the data support the conclusions?

Reviewer #2: Yes

3. Has the statistical analysis been performed appropriately and rigorously? 

Reviewer #2: Yes

4. Have the authors made all data underlying the findings in their manuscript fully available?

Reviewer #2: Yes

5. Is the manuscript presented in an intelligible fashion and written in standard English?

Reviewer #2: Yes

6. Review Comments to the Author

Reviewer #2: I have carefully read the edited version of the manuscript. The authors responded satisfactorily to all my comments. The manuscript thus revised may now be susceptible to publication.

7. PLOS authors have the option to publish the peer review history of their article (what does this mean?). If published, this will include your full peer review and any attached files.

Reviewer #2: No

---

## [Editor Report · Acceptance letter]

10 Jul 2024

PONE-D-23-38938R1 

PLOS ONE

Dear Dr. Woodfield, 

I'm pleased to inform you that your manuscript has been deemed suitable for publication in PLOS ONE. Congratulations! Your manuscript is now being handed over to our production team.

Kind regards, 

on behalf of

Dr. Alvan-Emeka K. Ukachukwu 

Academic Editor

PLOS ONE